# Sub-ppb Methane Detection via EMD–Wavelet Adaptive Thresholding in Wavelength Modulation TDLAS: A Hybrid Denoising Approach for Trace Gas Sensing

**DOI:** 10.3390/s25165167

**Published:** 2025-08-20

**Authors:** Tong Mu, Xing Tian, Peiren Ni, Shichao Chen, Yanan Cao, Gang Cheng

**Affiliations:** 1School of Artificial Intelligence, Anhui University of Science and Technology, Huainan 232001, China; 15178219810@163.com (T.M.); csc18856683605@163.com (S.C.); 2State Key Laboratory of Digital Intelligent Technology for Unmanned Coal Mining, Anhui University of Science & Technology, Huainan 232001, China; cynpf@mail.ustc.edu.cn (Y.C.); gang740@126.com (G.C.); 3Beijing Research Institute of Telemetry, Beijing 100076, China

**Keywords:** WM-TDLAS, second harmonic signals, EMD, wavelet adaptive thresholding, detection limit

## Abstract

Wavelength modulation-tunable diode laser absorption spectroscopy (WM-TDLAS) is a critical tool for gas detection. However, noise in second harmonic signals degrades detection performance. This study presents a hybrid denoising algorithm combining Empirical Mode Decomposition (EMD) and wavelet adaptive thresholding to enhance WM-TDLAS performance. The algorithm decomposes raw signals into intrinsic mode functions (IMFs) via EMD, selectively denoises high-frequency IMFs using wavelet thresholding, and reconstructs the signal while preserving spectral features. Simulation and experimental validation using the CH_4_ absorption spectrum at 1654 nm demonstrate that the system achieves a threefold improvement in detection precision (0.1181 ppm). Allan variance analysis revealed that the detection capability of the system was significantly enhanced, with the minimum detection limit (MDL) drastically reduced from 2.31 ppb to 0.53 ppb at 230 s integration time. This approach enhances WM-TDLAS performance without hardware modification, offering significant potential for environmental monitoring and industrial safety applications.

## 1. Introduction

Trace gases detection is of crucial importance in fields such as atmospheric science, chemistry, and biology. Detection methods for trace gases include spectroscopic techniques such as direct absorption spectroscopy (DAS), photoacoustic spectroscopy (PAS), Raman spectroscopy, wavelength-modulated tunable diode laser absorption spectroscopy (WM-TDLAS), micro fiber photoacoustic spectrometer (FPAS), light-induced thermoelastic spectroscopy (LITES), and surface-enhanced Raman spectroscopy (SERS) [1,2,3,4]. Among these, Wavelength Modulation-Tunable Diode Laser Absorption Spectroscopy (WM-TDLAS) has emerged as a cornerstone technology for trace gas detection, renowned for its high sensitivity and selectivity in applications ranging from environmental monitoring to industrial safety [5,6,7,8,9]. By leveraging the wavelength-dependent absorption of gas molecules, this technique extracts concentration information through the second harmonic signal (2f), which is particularly robust against laser intensity fluctuations [10]. However, practical implementations are often hindered by non-linear and non-stationary noise sources, including electronic noise, environmental perturbations, and laser frequency instability [11,12,13]. Such noise degrades the signal-to-noise ratio (SNR), compromising the accuracy of gas concentration measurements and limiting the detection sensitivity to sub-ppb levels [14]. For instance, methane leaks in natural gas pipelines or coal mines require detection limits below 1 ppb to ensure safety and environmental compliance, yet existing methods struggle to achieve this threshold consistently [15].

Traditional denoising approaches, such as Fourier transform-based filtering and conventional wavelet thresholding, struggle to effectively suppress noise in WM-TDLAS signals due to their inherent assumptions of linearity and stationarity [16,17]. While Empirical Mode Decomposition (EMD) has shown promise in adaptively decomposing non-linear signals into intrinsic mode functions (IMFs), it often introduces mode mixing and may discard valuable high-frequency features [18,19,20]. Conversely, wavelet transform provides multi-resolution analysis but relies on fixed basis functions and heuristic threshold selection, which may not optimally adapt to the complex noise characteristics of WM-TDLAS [21]. These limitations highlight the need for a hybrid approach that combines the strengths of EMD and wavelet transform while mitigating their individual drawbacks.

To address these challenges, this study introduces a hybrid denoising framework that combines the adaptive decomposition capabilities of EMD with the multi-scale precision of wavelet transform. The proposed EMD–Wavelet Adaptive Thresholding approach dynamically identifies and preserves spectral features embedded in high-frequency IMFs while suppressing noise through data-driven thresholding. By integrating these complementary techniques, the method achieves superior noise reduction compared to traditional approaches, as validated through simulations and experimental measurements of CH_4_ absorption at 1654 nm. Specifically, the algorithm reduces the minimum detection limit from 2.31 ppb to 0.53 ppb at an integration time of 230 s, representing a 77% improvement [22]. This work not only advances signal processing for spectroscopic applications but also enables WM-TDLAS systems to achieve sub-ppb detection limits, opening new possibilities for real-time monitoring of methane leaks in critical infrastructure and atmospheric environments. The results contribute to the development of next-generation gas sensors that meet the stringent requirements of safety-critical and environmentally sensitive scenarios.

## 2. Principles and Approaches

### 2.1. Empirical Mode Decomposition (EMD)

Empirical Mode Decomposition (EMD) is a data-driven signal processing technique that adaptively decomposes non-linear and non-stationary signals into a set of intrinsic mode functions (IMFs) and a residual term [22,23]. Each IMF represents a local oscillatory mode with well-defined amplitude and frequency, while the residual captures the signal’s long-term trend. The decomposition process, rooted in the Hilbert–Huang Transform (HHT), involves iteratively extracting IMFs by identifying local extrema, constructing upper and lower envelopes via cubic spline interpolation, and subtracting their mean from the original signal. This ensures each IMF satisfies two conditions: (1) the number of extrema and zero-crossings must either be equal or differ by at most one; and (2) the mean of the upper and lower envelopes must be zero. These criteria guarantee orthogonality between IMFs and preserve the signal’s physical interpretability.

For WM-TDLAS signals, EMD is particularly advantageous due to its ability to adaptively isolate noise-dominated high-frequency components from the underlying harmonic signal. Specifically, the algorithm decomposes the raw 2f signal into IMFs ordered by decreasing frequency. High-frequency IMFs (e.g., IMF1–IMF3) are typically associated with noise, while lower-frequency IMFs retain spectral features. However, traditional EMD-based denoising often discards high-frequency IMFs entirely, potentially removing valuable transient signal components. To address this, our framework introduces a cross-correlation analysis to quantify the contribution of each IMF to the original signal, enabling selective processing of noise-contaminated IMFs.

### 2.2. Wavelet Adaptive Thresholding

Wavelet threshold denoising can be categorized into three primary steps: decomposition, thresholding, and reconstruction [24]. As illustrated in Figure 1, the process begins with the decomposition of the signal into wavelet coefficients at multiple resolution levels. This is followed by the application of a thresholding function to suppress noise-dominated coefficients while preserving significant signal components. Finally, the denoised signal is reconstructed by performing an inverse wavelet transform on the thresholded coefficients. A detailed schematic of the wavelet threshold denoising process is provided in Figure 1.

Thresholding involves deciding whether a coefficient should be set to zero based on a predetermined threshold value. Thresholding techniques are generally divided into hard and soft thresholding. Hard thresholding preserves coefficients above the threshold and sets the others to zero, while soft thresholding shrinks coefficients above the threshold toward zero, resulting in a smoother denoising effect. Threshold estimation methods include heuristic, fixed, unbiased likelihood, and min–max threshold estimation [25]. Among these, min–max and unbiased likelihood thresholding are well-established traditional methods, particularly effective for signals exhibiting high signal-to-noise ratios (SNR). To enhance performance across a wider range of SNR conditions, this paper employs an adaptive thresholding method based on unbiased likelihood threshold estimation. Specifically, we preprocess each element *s* to generate a new sequence *f*(*k*), defined as(1)$$f\left(k\right)={\left[sort(\left|s\right|)\right]}^{2}$$

In the question, *k* = 1, 2, 3, …, *N*, $sort\left(\left|s\right|\right)$ represent the ascending order.

Assume that *f*(*k*) is the threshold; then, the threshold risk vector is(2)$$R(k)=\left[N-2k+\sum_{i=1}^{k}f\left(j\right)+\left(N-k\right)f\left(N-k\right)\right]/N$$
where *R*(*k*) takes the minimum value, and the corresponding value of $k_{\min}$ is the threshold.(3)$$\lambda_{k}=\sqrt{f\left(k_{\min}\right)}$$

### 2.3. EMD–Wavelet Adaptive Thresholding Denoising Framework

To efficiently remove noise from second harmonic signals, we propose a denoising method that integrates Empirical Mode Decomposition (EMD) and wavelet adaptive thresholding, as illustrated in Figure 2. The process begins with the decomposition of the noisy signal using EMD, which generates a set of intrinsic mode functions (IMFs) arranged in descending order of frequency. This decomposition separates the noisy signal into multiple IMF components, each corresponding to distinct frequency bands. Subsequently, the cross-correlation coefficients between each IMF and the original signal are computed. IMFs with cross-correlation coefficients below a predefined threshold are identified for further processing. While these IMFs predominantly contain noise, they may also retain valuable signal features, making their complete dismissal undesirable.

To extract the useful information embedded within these IMFs, a wavelet adaptive thresholding technique is employed. This involves calculating the wavelet coefficients of the selected IMFs and applying an adaptive threshold to distinguish between noise and signal components. The thresholding process is designed to preserve critical signal features while effectively suppressing noise. Following this, wavelet reconstruction is performed to eliminate the noise, yielding a denoised signal sequence. Finally, the denoised IMFs are combined with the remaining retained IMFs to reconstruct the final denoised signal, ensuring a high-fidelity representation of the original signal with significantly reduced noise.

## 3. Simulation Analysis

### 3.1. Simulation Model

To evaluate the denoising performance of the combined Empirical Mode Decomposition (EMD) and wavelet adaptive thresholding method, a comprehensive simulation analysis was conducted. The second harmonic signal of CH_4_ at 6046.96 cm^−1^ was simulated under conditions consistent with the experimental setup described in this study, an atmospheric pressure of 1 atm, a temperature of 295 K, an absorption path length of 29.37 m, and a central wavelength of 1654 nm. The simulation generated 2000 data points, with white noise added to replicate the noise typically encountered during the harmonic signal measurement process. The ideal second harmonic signal, obtained through fitting, is illustrated in Figure 3a, while the corresponding noisy second harmonic signal is shown in Figure 3b.

To quantitatively assess the denoising performance, this study employs three key metrics: signal-to-noise ratio (SNR), root mean square error (RMSE), and correlation coefficient (CC) [26,27,28]. The SNR provides a measure of the noise reduction achieved by the denoising process, while the RMSE quantifies the deviation between the denoised signal and the original noise-free signal. The CC evaluates the degree of similarity between the denoised second harmonic signal and the original signal, offering insight into the preservation of signal integrity. The signal-to-noise ratio (SNR) is defined as follows:(4)$$\beta_{SNR}=10\mathrm{lg}\frac{\mathrm{sum}\left\{{\left[S_{\mathrm{signal}}{\left(x\right)}_{2f}\right]}^{2}\right\}}{\mathrm{sum}\left\{{\left[S_{\mathrm{signal}}{\left(x\right)}_{2f}-S_{\mathrm{signal}}{\left(x\right)}_{\mathrm{denoised}-2f}\right]}^{2}\right\}}$$
where $\beta_{SNR}$ represents the signal-to-noise ratio (SNR), $S_{\mathrm{signal}}{\left(x\right)}_{2f}$ is the original second harmonic signal, $S_{\mathrm{signal}}{\left(x\right)}_{\mathrm{denoised}-2f}$ is the denoised second harmonic signal.(5)$$\alpha_{\mathrm{RMSE}}=\sqrt{\sum_{x=1}^{N^{\prime }}\frac{{\left[S_{\mathrm{signal}}{\left(x\right)}_{2f}-S_{\mathrm{signal}}{\left(x\right)}_{\mathrm{denoised}-2f}\right]}^{2}}{N^{\prime }}}$$

In the equation, $\alpha_{\mathrm{RMSE}}$ represents the root mean square error, and $N^{\prime }$ is the signal length.

### 3.2. Performance Analysis of EMD–Wavelet Adaptive Thresholding

The EMD–Wavelet Adaptive Thresholding denoising algorithm is applied to process the noisy signal. First, adhering to the IMF decomposition conditions, the second harmonic signal is decomposed layer by layer using the EMD algorithm, as illustrated in Figure 4. The original second harmonic signal is decomposed into nine intrinsic mode functions (IMFs), labeled IMF1 through IMF9, along with a residual component (Res), arranged in order of decreasing frequency. The waveforms of IMF1–IMF3 span the entire time-sampling axis without exhibiting attenuation, indicating that they primarily correspond to high-frequency noise. In traditional EMD low-pass denoising methods, the IMF1–IMF3 components are typically discarded, and the remaining IMFs are used to reconstruct the denoised signal.

However, the observed discrepancy between the denoised signal and the original signal suggests that some useful feature information within the IMF1–IMF3 components has been erroneously identified as noise and removed, resulting in signal distortion. To more accurately distinguish between noise-contaminated IMF components and those containing useful signal features, the correlation coefficient is employed to evaluate the relationship between each IMF component and the original signal. This approach enables the precise identification of noisy IMF components, which are then subjected to further processing using the wavelet adaptive threshold denoising method. The results of this analysis are summarized in Table 1, where IMF1–IMF3 components with correlation coefficients below 0.1 are classified as noise-contaminated and targeted for denoising.

Following the correlation coefficient analysis, it was determined that IMF1–IMF3, obtained from the EMD, primarily consist of high-frequency noise components but also retain valuable feature information. To avoid the inadvertent loss of these useful features—a common issue with traditional EMD low-pass methods—a wavelet adaptive thresholding denoising technique was applied to IMF1–IMF3. This method effectively separates noise from the underlying signal, enabling the extraction of valid feature information that was previously obscured by noise, as demonstrated in Figure 5.

Subsequently, the processed IMF1–IMF3 components are combined with the remaining retained IMF components to reconstruct the denoised signal, as shown by the blue line in Figure 6. From Figure 6, it is evident that noise is effectively suppressed, resulting in a significantly smoother signal with enhanced denoising performance, particularly at the peak regions, while preserving potentially useful information. This outcome confirms that the valuable feature information within IMF1–IMF3 has been successfully extracted and retained. Additionally, Figure 6 provides a comparison between the denoised signal obtained using the EMD–Wavelet Adaptive Thresholding denoising method (blue line) and the ideal second harmonic signal (red dots). The performance metrics demonstrate an RMSE of 0.0067%, an SNR of 29.7885, and a CC of 99.9503%, underscoring the high fidelity and effectiveness of the proposed denoising method.

### 3.3. Comparison Analysis with Other Methods

To further validate the effectiveness of the EMD–Wavelet Adaptive Thresholding denoising algorithm, this study compares it with several traditional denoising methods, including EMD denoising, wavelet hard-thresholding denoising, wavelet soft-thresholding denoising, and wavelet adaptive thresholding denoising [29,30,31,32]. The denoising performance of each algorithm was evaluated by comparing the denoised second harmonic signals with the original signals, as illustrated in Table 2. Quantitative evaluation metrics for each algorithm’s performance are summarized in Table 2. The results demonstrate that the EMD–Wavelet Adaptive Thresholding denoising method outperforms the other four methods, with the denoised second harmonic curve showing significant improvement. These findings indicate that the second harmonic signal processed by the proposed method closely approximates the pure signal curve, with better preservation of peak positions, underscoring the superior denoising performance of the EMD–Wavelet Adaptive Thresholding denoising approach.

## 4. Experiments

### 4.1. Experimental System

To validate the algorithm’s performance with experimental data, we developed a wavelength modulation-tunable diode laser absorption spectroscopy (WM-TDLAS) system for methane (CH_4_) detection, as illustrated in Figure 7. The system was designed to precisely tune the laser output frequency to the peak absorption wavelength of methane. A distributed feedback (DFB) diode laser emitting at 1654 nm served as the light source. The laser’s current and temperature were regulated using a commercial diode laser controller (Model LDC 501, Stanford Research Systems, US). Wavelength tuning was achieved through coarse adjustments via temperature control and fine adjustments via current modulation. A linear wavelength scan was implemented by applying a voltage ramp from a function generator (DG4162, RIGOL, China) to the laser diode current, enabling the laser wavelength to sweep across the absorption line at a rate of 1 Hz. Additionally, a 6 kHz sinusoidal modulation signal from a lock-in amplifier (Model SR 830 DSP, Stanford Research Systems, US) was superimposed onto the laser current using a custom-built adder circuit to facilitate wavelength modulation. The laser beam was collimated using a fiber-coupled collimator (focal length ~4.8 mm) and directed into a compact dense-pattern multi-pass cell (DP-MPC). The DP-MPC, comprising two 2-inch silver-coated concave spherical mirrors separated by 12 cm, provided an effective optical path length of 29.37 m through 243 reflections [33,34]. The output beam was focused onto a photodetector (PDA20CS-EC, Thorlabs, US) using a 50 mm focal length lens. The photodetector signal was demodulated by the lock-in amplifier, digitized using a data acquisition card (NI USB-6210, National Instruments, US), and processed via a LabVIEW interface on a laptop. Simultaneously, the photodetector signal was monitored to ensure laser power stability throughout the measurements.

### 4.2. Experimental Results and Discussion

In the experiment, five groups of CH_4_ gas samples with concentrations of 10 ppm, 20 ppm, 30 ppm, 40 ppm, and 50 ppm were prepared using the gas distribution system. The linear response of the second harmonic (2f) signal amplitude to CH_4_ concentrations was measured, as shown in Figure 8a. To enhance signal quality and reduce noise, an Empirical Mode Decomposition (EMD)–Wavelet Adaptive Threshold denoising method was applied during the data processing phase. This advanced technique effectively filters the raw signal, as demonstrated in Figure 8b, which displays the denoised second harmonic (2f) signal. A comparison between Figure 8a (original signal) and Figure 8b (denoised signal) reveals a significant reduction in overall noise, resulting in a smoother curve that more clearly reveals the gas concentration information. The denoising process not only improves the signal-to-noise ratio (SNR) but also enhances the accuracy of gas concentration extraction.

Figure 9 illustrates the correlation between the amplitude of the second harmonic (2f) signal, processed using the EMD–Wavelet Adaptive Thresholding denoising method, and the concentration of CH_4_. The linear response of the CH_4_ signal to CH_4_ concentration variations was confirmed by fitting the data with a linear slope. The R-squared value of the linear fit is greater than 0.99, demonstrating a strong linear relationship between the 2f signal and the CH_4_ concentration.

### 4.3. Performance Evaluation

System stability represents a crucial factor determining the sensitivity of gas detection systems. To quantitatively assess the long-term stability of our system, we performed continuous gas monitoring experiments under well-controlled laboratory conditions. A calibrated 30 ppm methane (CH_4_) gas sample was introduced into the multi-pass absorption cell, and the concentration was continuously monitored over a 60 min period with a temporal resolution of 1 s. Through application of the spectroscopic inversion algorithm, a total of 3600 concentration data points were acquired. Figure 10 presents the temporal evolution of the measured CH_4_ concentration measured before and after EMD–Wavelet Adaptive Thresholding denoising signal, demonstrating the system’s stability characteristics. The experimental results not only provide a quantitative evaluation of the system’s long-term stability but also offer valuable insights into its potential performance in practical field applications.

To systematically evaluate the measurement accuracy enhancement achieved through the EMD–Wavelet Adaptive Thresholding denoising method in the WM-TDLAS system, we conducted a comprehensive statistical analysis of the measurement data distribution, as illustrated in Figure 11. The comparative histogram analysis demonstrates a remarkable improvement in data quality following the denoising process. The raw data distribution prior to denoising exhibits substantial dispersion with a relatively broad profile, suggesting significant measurement variability. In contrast, the post-denoising histogram reveals a pronounced, narrow Gaussian distribution centered precisely at the target concentration of 30 ppm, indicating enhanced measurement consistency. Quantitative analysis based on Gaussian fitting demonstrates that the full-width at half-maximum (FWHM) of the concentration distribution was reduced from 0.3547 ppm to 0.1181 ppm after denoising implementation. This corresponds to a 66.7% reduction in FWHM, equivalent to a threefold improvement in measurement precision. These results conclusively demonstrate the efficacy of the EMD–Wavelet Adaptive Thresholding denoising method in significantly enhancing the measurement accuracy and reliability of WM-TDLAS systems.

The minimum detection limit (MDL) represents a crucial performance parameter for gas detection systems, serving as a key indicator of their sensitivity and precision. To comprehensively assess the system’s MDL and its correlation with averaging time, we conducted an Allan variance analysis. Figure 12 presents the Allan deviation plots obtained from 3600 methane (CH_4_) concentration measurements, comparing the system’s performance characteristics before and after implementation of the EMD–Wavelet Adaptive Thresholding denoising algorithm. The analytical results reveal a remarkable enhancement in the system’s detection capability, with the MDL showing a significant reduction from 2.31 ppb to 0.53 ppb. This substantial improvement, representing a 77% reduction in detection limit, clearly demonstrates the efficacy of the proposed denoising technique in suppressing noise and enhancing detection sensitivity. Furthermore, the observed improvements in both MDL and optimal averaging time collectively indicate that the developed method effectively enhances the system’s detection accuracy while simultaneously improving its long-term stability. These performance enhancements make the proposed system particularly suitable for demanding applications requiring high-precision trace gas detection, such as environmental monitoring and industrial process control.

## 5. Conclusions

To eliminate noise interference in second harmonic signals for gas concentration detection via near-infrared wavelength modulation spectroscopy (WMS), we propose a novel EMD–Wavelet Adaptive Threshold denoising algorithm. Comparative simulations with four established denoising techniques confirm its superior performance, boosting the signal-to-noise ratio (SNR) from 7.5182 to 29.7885. Implementation in a wavelength-modulated tunable diode laser absorption spectroscopy (WM-TDLAS) system for CH_4_ detection yielded substantial improvements: detection precision tripled to 0.1181 ppm. Allan deviation analysis revealed a minimum detection limit (MDL) of 0.53 ppb at an optimal integration time of 230 s—all without hardware modifications to the existing TDLAS architecture. This algorithm advances WMS-based gas detection by enhancing sensitivity and stability, offering broad applicability for upgrading TDLAS systems in both laboratory and field settings. It contributes to the ongoing refinement of precision gas detection methodologies in the near-infrared spectrum.

## Figures and Tables

**Figure 1 sensors-25-05167-f001:**
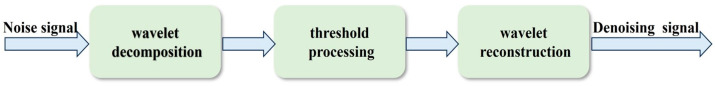
The wavelet thresholding denoising process.

**Figure 2 sensors-25-05167-f002:**
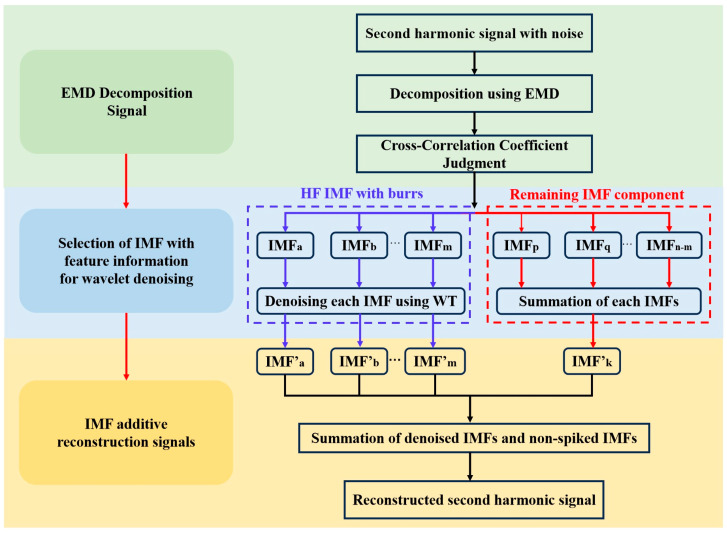
EMD–Wavelet Adaptive Thresholding denoising framework.

**Figure 3 sensors-25-05167-f003:**
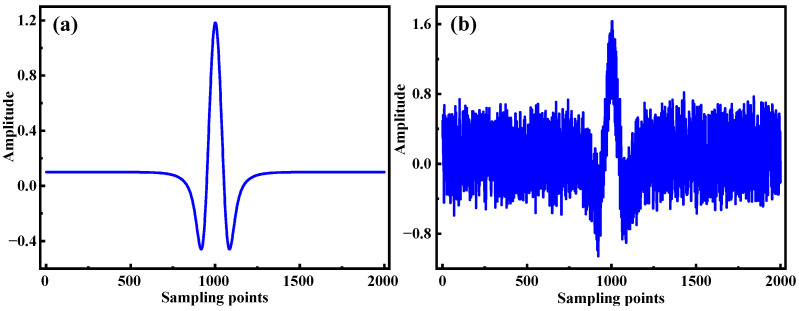
Simulated CH_4_ second harmonic signal. (**a**) Original second harmonic signal; (**b**) second harmonic signal with white noise.

**Figure 4 sensors-25-05167-f004:**
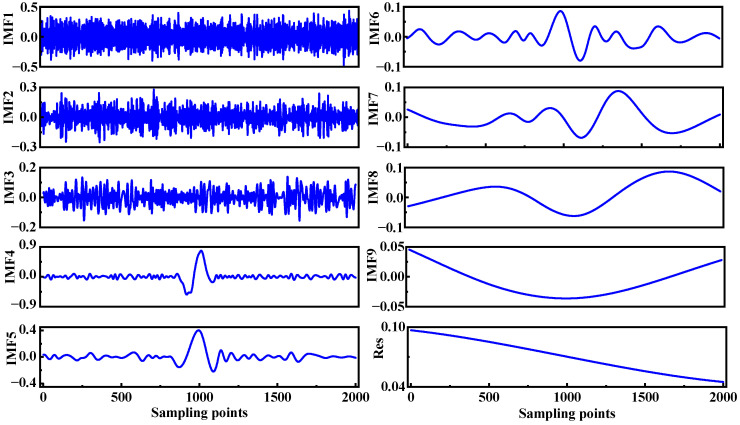
The IMF components and a residual component of the second harmonic signal.

**Figure 5 sensors-25-05167-f005:**
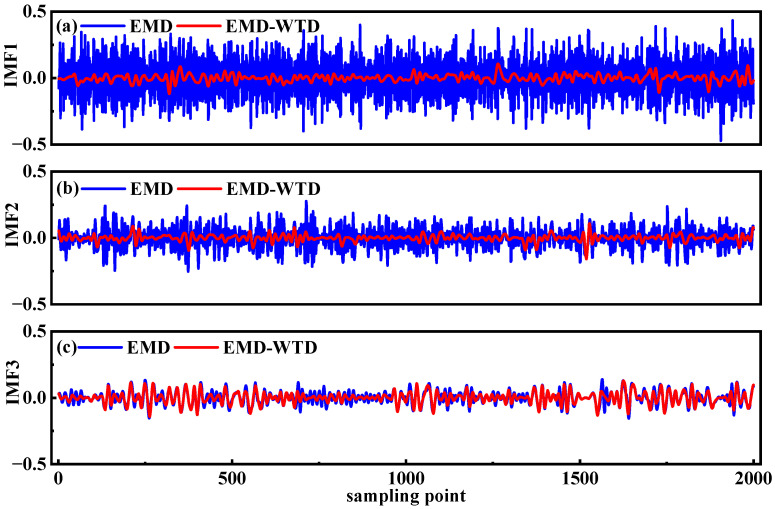
Comparison of IMF components after EMD–Wavelet Adaptive Thresholding denoising. (**a**–**c**) IMF1–IMF3.

**Figure 6 sensors-25-05167-f006:**
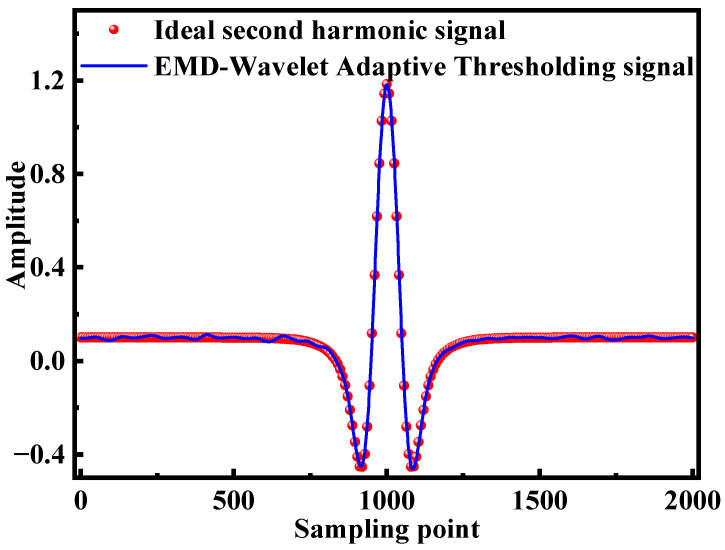
Comparison between the EMD–Wavelet Adaptive Threshold denoising signal and the ideal second harmonic signal.

**Figure 7 sensors-25-05167-f007:**
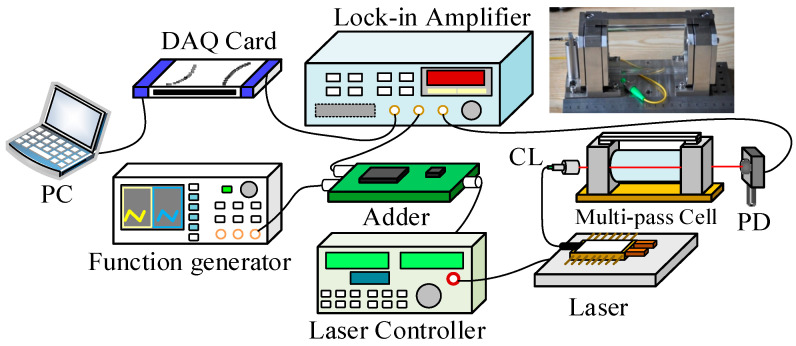
Schematic diagram of the experimental system.

**Figure 8 sensors-25-05167-f008:**
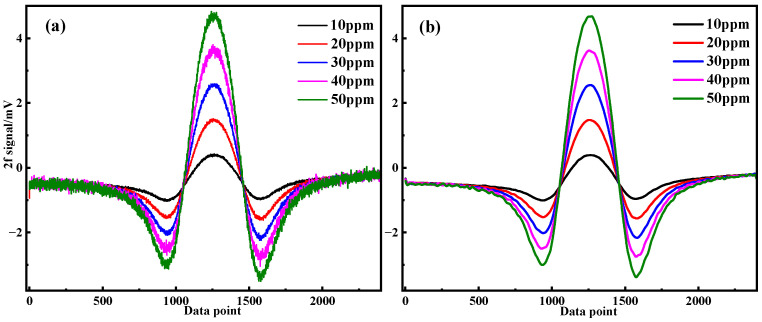
The 2f signal waveforms for different CH_4_ concentrations. (**a**) Original signal; (**b**) EMD–Wavelet Adaptive Thresholding denoising signal.

**Figure 9 sensors-25-05167-f009:**
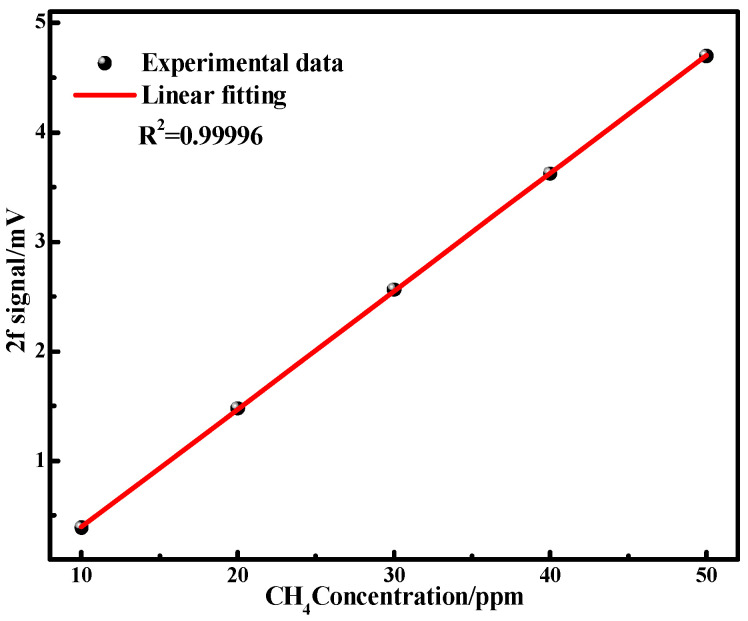
The 2f signal amplitude of CH_4_ and fitting curve as a function of CH_4_ concentrations.

**Figure 10 sensors-25-05167-f010:**
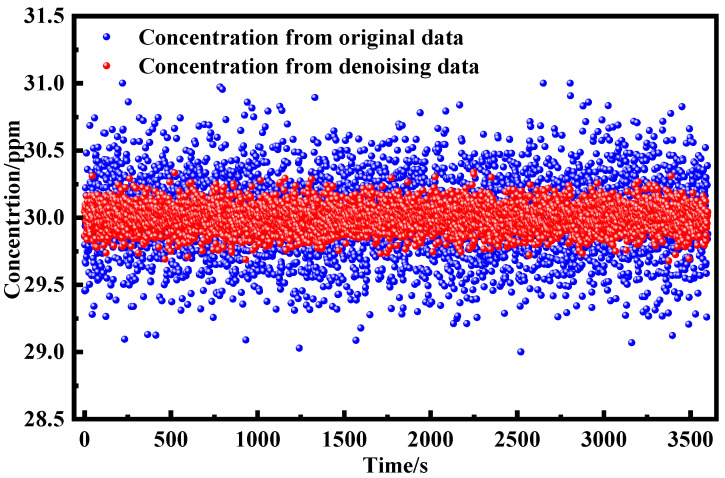
Continuous concentration test results before and after EMD–Wavelet Adaptive Thresholding denoising.

**Figure 11 sensors-25-05167-f011:**
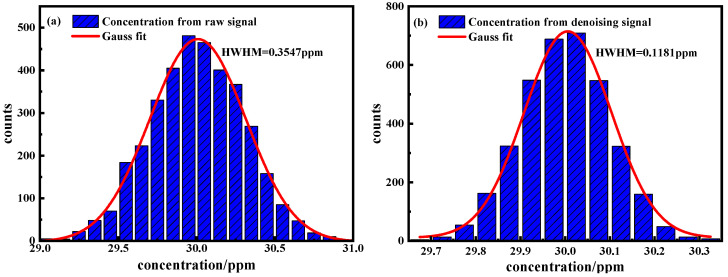
Histogram plot obtained from time-series measurements of 30 ppm CH_4_. (**a**) Original signal; (**b**) EMD–Wavelet Adaptive Thresholding denoising signal.

**Figure 12 sensors-25-05167-f012:**
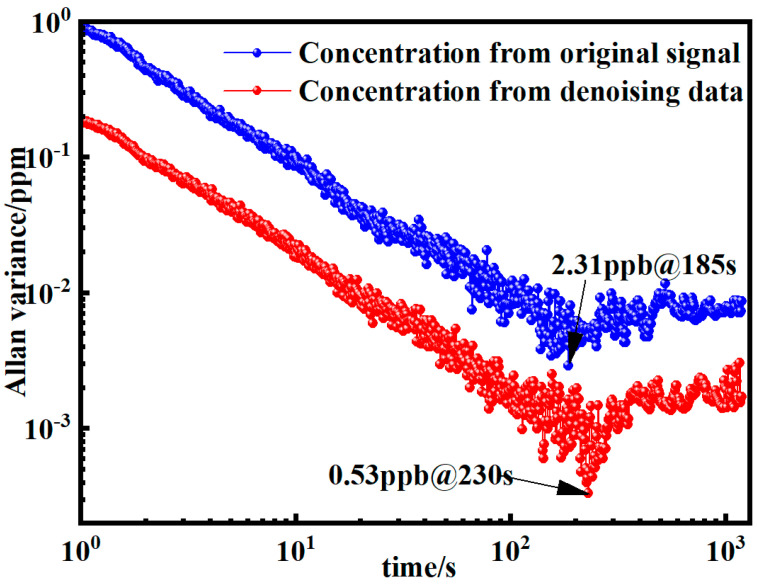
Allan deviation analysis before and after EMD–Wavelet Adaptive Thresholding denoising.

**Table 1 sensors-25-05167-t001:** The correlation coefficient between the IMF components and the original signal.

IMFs	Correlation Coefficient	IMFs	Correlation Coefficient
IMF1	0.0082	IMF6	0.5755
IMF2	0.0042	IMF7	0.1575
IMF3	0.0155	IMF8	0.1216
IMF4	0.8454	IMF9	0.1491
IMF5	0.7925	Res	0.000594

**Table 2 sensors-25-05167-t002:** Comparison of denoising effects with different algorithms.

Denoising Method	RMSE/%	SNR	CC/%
Noisy signal	0.7105	7.5182	56.14
EMD	0.0241	23.4028	99.7016
Wavelet hard thresholding	0.0194	26.8846	99.8713
Wavelet soft thresholding	0.0107	27.4227	99.9037
Wavelet adaptive thresholding	0.0218	24.8721	99.7713
EMD–Wavelet Adaptive Thresholding	0.0067	29.7885	99.9503

## Data Availability

The original contributions presented in this study are included in the article. Further inquiries can be directed to the corresponding authors.

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
