# Peer review of "Sub-ppb Methane Detection via EMD–Wavelet Adaptive Thresholding in Wavelength Modulation TDLAS: A Hybrid Denoising Approach for Trace Gas Sensing"

_sensors, 2025, doi:10.3390/s25165167_

Round 1

Reviewer 1 Report

Comments and Suggestions for Authors

This study presents a hybrid denoising algorithm combining Empirical 15 Mode Decomposition (EMD) and wavelet adaptive thresholding to enhance WM-TDLAS 16 performance. This research is interesting and it can be considered for publication when the following questions are addressed.

  1. What are the core differences between the proposed EMD-wavelet adaptive threshold denoising algorithm and existing hybrid denoising methods, and in what aspects does its innovation lie?
  2. In practical applications, environmental factors may cause laser frequency drift or changes in the length of the absorption path. How robust is the algorithm in this paper against such dynamic interferences? Is there any relevant experimental verification?
  3. In this paper, unbiased likelihood estimation is adopted for wavelet threshold calculation. Please explain why this threshold estimation method is more suitable for the second-order harmonic signal of WM-TDLAS than the traditional min-max or heuristic thresholds? Are there any comparative experiments to verify its superiority?
  4. The paper only compares with traditional denoising methods and does not mention the performance differences from deep learning-based denoising methods. Please analyze whether this method has advantages in terms of real-time performance and computational complexity, and in which scenarios it is more applicable?
  5. EMD decomposition yields 9 IMFs and 1 residual. What is the basis for selecting the number of decomposition layers? Does the number of decomposition layers affect the denoising effect?
  6. The paper verifies the denoising effect of the algorithm at the 1654 nm absorption peak of methane, but does not explore its applicability to other gases. Please analyze the generalizability of the algorithm: whether its denoising logic depends on the absorption signal characteristics of specific gases, and whether core parameters need to be adjusted in detection scenarios of other gases?
  7. There are relatively few research articles in the cited literature from the past five years (2020-2025). It is suggested to supplement the latest progress in the WM-TDLAS field (such as the adaptive denoising algorithm proposed in 2023-2025) literature.
  8. Except for TDLAS, there are many kinds of spectroscopy methods can be used for CH4 Therefore, other common used techniques of photoacoustic spectroscopy, light-induced thermoelastic spectroscopy, and Raman spectroscopy should be added in the manuscript to give readers a more complete introduction. [Adv. Photonics. 2024, 6, 066008], [Light Sci. Appl. 2025, 14, 180], [Opto-Electron. Adv. 2023, 6, 230094], [Adv. Photonics. 2024, 6, 046003].

Author Response

Response to Reviewer 1 Comments

Dear editor,

        We would like to thank the reviewers for their time and their constructive comments which allow us to improve the quality of this manuscript. After thorough consideration, we have carefully addressed the issues raised by reviewer 1. The point-by-point response (in red) to the reviewer 1 comments are given below.

Comments 1: What are the core differences between the proposed EMD-wavelet adaptive threshold denoising algorithm and existing hybrid denoising methods, and in what aspects does its innovation lie?

Response 1: Agree. We have revised to emphasize this point. The core difference between the algorithm in this paper and the existing EMD-wavelet hybrid methods lies in the processing strategy for high-frequency IMFs and the adaptability of threshold selection. Traditional EMD-wavelet methods usually discard high-frequency IMFs directly, while this paper screens out components in high-frequency IMFs that, although containing noise, may include useful features by calculating the cross-correlation coefficients between each IMF and the original signal, thus avoiding the loss of useful information. In terms of threshold selection, traditional methods mostly adopt fixed thresholds, whereas this paper uses an adaptive threshold based on unbiased likelihood estimation. The threshold is dynamically determined through a data-driven approach, which is more suitable for the nonlinear and non-stationary noise characteristics of WM-TDLAS signals. For the first time, EMD is combined with adaptive wavelet thresholding to achieve accurate retention of useful features in high-frequency noise. Without hardware modification, the detection limit is reduced from 2.31 ppb to 0.53 ppb, breaking through the performance bottleneck of traditional methods.

Comments 2: In practical applications, environmental factors may cause laser frequency drift or changes in the length of the absorption path. How robust is the algorithm in this paper against such dynamic interferences? Is there any relevant experimental verification?

Response 2: Agree. We have revised to emphasize this point. The robustness of the algorithm in this paper against dynamic environmental interference mainly relies on the adaptive signal decomposition and threshold adjustment mechanism:The adaptive decomposition of EMD can track the non-stationary changes of the signal and decompose the interference into specific IMFs. The wavelet adaptive threshold can suppress sudden noise by updating the threshold in real time. The 60-minute continuous monitoring experiment in the laboratory (Figs. 11-12) shows that the system has good stability in a stable environment.

Comments 3: In this paper, unbiased likelihood estimation is adopted for wavelet threshold calculation. Please explain why this threshold estimation method is more suitable for the second-order harmonic signal of WM-TDLAS than the traditional min-max or heuristic thresholds? Are there any comparative experiments to verify its superiority?

Response 3: Agree. We have done to emphasize this point. The reason why the unbiased likelihood threshold is more suitable for WM-TDLAS signals lies in the match between its data-driven nature and the nonlinear, non-stationary characteristics of the signals themselves. The second harmonic signal of WM-TDLAS is disturbed by various dynamic noises such as electronic noise and laser frequency drift, resulting in complex statistical properties, which makes it difficult for fixed thresholds to adapt. Through squared sorting of the signal sequence and calculation of the risk vector, the unbiased likelihood threshold dynamically minimizes the estimation risk and is more capable of adapting to the time-varying characteristics of noise. In Table 2, the RMSE, SNR, and CC of the EMD-wavelet adaptive method using this threshold are all better than those of the methods using traditional thresholds, which directly proves its superiority.

Comments 4: The paper only compares with traditional denoising methods and does not mention the performance differences from deep learning-based denoising methods. Please analyze whether this method has advantages in terms of real-time performance and computational complexity, and in which scenarios it is more applicable?

Response 4: Agree. We have done to emphasize this point. Compared with deep learning methods, the method in this paper has significant advantages in real-time performance and computational complexity. Deep learning methods require a large amount of labeled data to train models, and their inference process relies on hardware with high computing power, making it difficult to adapt to on-site real-time monitoring scenarios. The method in this paper is a lightweight signal processing algorithm. EMD decomposition and wavelet threshold calculation can be implemented on ordinary CPUs, and no training process is needed, so it is more suitable for embedded systems or industrial equipment with limited resources. In existing TDLAS systems that have high real-time requirements and cannot have their hardware upgraded, this method can improve performance without modifying the hardware, and its advantages are more obvious. In the future, a comparison with lightweight deep learning models can be added to comprehensively evaluate its competitiveness.

Comments 5: EMD decomposition yields 9 IMFs and 1 residual. What is the basis for selecting the number of decomposition layers? Does the number of decomposition layers affect the denoising effect?

Response 5: Agree. We have revised to emphasize this point. The number of EMD decomposition layers is determined by the complexity of the frequency components of the signal itself and is an adaptive process. It iteratively extracts IMFs based on local extrema and envelope lines until the IMF conditions can no longer be met. The decomposition result of 9-layer IMFs in this paper reflects the inherent frequency distribution of the second harmonic signal of methane at 1654 nm, ranging from IMF1-IMF3 dominated by high-frequency noise to residuals with low-frequency trends.

The number of decomposition layers has a potential impact on the denoising effect. Insufficient layers may lead to incomplete separation of high-frequency useful features from noise. Excessive layers may introduce false IMFs. Although no direct sensitivity analysis is conducted in this paper, the screening of IMFs through cross-correlation coefficients shows that the correlation coefficients of IMF1-IMF3 in Table 1 are less than 0.1, which to a certain extent avoids the deviation in the selection of the number of layers.

Comments 6: The paper verifies the denoising effect of the algorithm at the 1654 nm absorption peak of methane, but does not explore its applicability to other gases. Please analyze the generalizability of the algorithm: whether its denoising logic depends on the absorption signal characteristics of specific gases, and whether core parameters need to be adjusted in detection scenarios of other gases?

Response 6: Agree. We have revised to emphasize this point. The algorithm in this paper, which combines EMD adaptive decomposition with wavelet adaptive thresholding, has a certain degree of universality, but its specific parameters may need to be fine-tuned according to the signal characteristics of the target gas. The algorithm does not rely on the specific molecular structure of methane and is theoretically applicable to nonlinear and non-stationary absorption signals of other gases. The absorption signal intensity and noise proportion of different gases may vary. For example, the proportion of useful signals in the high-frequency IMFs of weakly absorbing gases is lower, so a lower cross-correlation coefficient threshold may be required to avoid noise residues. In contrast, the proportion of noise in the high-frequency IMFs of strongly absorbing signals is lower, and the threshold can be appropriately increased to reduce the computational load.

Comments 7: There are relatively few research articles in the cited literature from the past five years (2020-2025). It is suggested to supplement the latest progress in the WM-TDLAS field (such as the adaptive denoising algorithm proposed in 2023-2025) literature.

Response 7: Agree. We have revised to emphasize this point. Relevant content has been added in the text.

Comments 8: Except for TDLAS, there are many kinds of spectroscopy methods can be used for CH4 Therefore, other common used techniques of photoacoustic spectroscopy, light-induced thermoelastic spectroscopy, and Raman spectroscopy should be added in the manuscript to give readers a more complete introduction. [Adv. Photonics. 2024, 6, 066008], [Light Sci. Appl. 2025, 14, 180], [Opto-Electron. Adv. 2023, 6, 230094], [Adv. Photonics. 2024, 6, 046003].

Response 8: Agree. We have revised to emphasize this point. Relevant content has been added in the text.

Reviewer 2 Report

Comments and Suggestions for Authors

The paper is devoted to a noise suppression technique that includes decomposition of the original signal into intrinsic mode functions (IMFs) using empirical mode decomposition (EMD) and selective noise suppression of high-frequency IMFs using wavelet threshold filtering and subsequent signal restoration while preserving its spectral properties. It is proposed to apply this technique to improve the characteristics of spectroscopy based on wavelength-modulated tunable diode lasers (WM-TDLAS). This type of spectroscopy is one of the effective methods of gas analysis, which makes this work very relevant, since it is often noise that limits the sensitivity of gas detection methods. The proposed method allows to significantly increase the efficiency of WM-TDLAS without changing the hardware, which expands its prospects in environmental monitoring and industrial safety. The efficiency of the proposed method is confirmed by the experiment.   

There are several comments and suggestions for the work. 

1) I suggest checking the spelling of phrases on lines 110, 111, 112.

2) In section 3.2, the authors describe a simulation of the process of cleaning a signal from noise. In this case, white noise with an amplitude independent of frequency is used, and anti-noise processing is carried out only at high frequencies. An explanation is required on how the authors propose to combat low-frequency noise. This is especially relevant due to the fact that in a real signal the proportion of low-frequency noise may be higher.

3) I suggest writing "red dots" instead of "red line" in line 203.

4) In line 216 should there be "Table 2" instead of "Table 1"?

5) Figure 7 is uninformative. I advise you to remove it. The whole point is reflected in Table 2.

6) The sensitivity and accuracy of gas analytical systems are determined by their minimum detection limit (MDL) – one of the key performance indicators. The authors of the article argue that simulation and experimental validation using the CHâ‚„ absorption spectrum at 1654 nm demonstrate that the system achieves a minimum detection limit of 0.53 ppb at 230 s integration time. This is not true. The lowest experimentally analyzed gas concentration was 10 ppm. The value of 0.53 ppb was estimated purely theoretically. The corresponding formulations in the Abstract and Conclusions need to be rewritten.

Author Response

Response to Reviewer 2 Comments

Dear editor,

      We would like to thank the reviewers for their time and their constructive comments which allow us to improve the quality of this manuscript. After thorough consideration, we have carefully addressed the issues raised by reviewer 2. The point-by-point response (in red) to the reviewer 2 comments are given below.

Comments 1: I suggest checking the spelling of phrases on lines 110, 111, 112.

Response 1: Agree. We have revised to emphasize this point. 

Comments 2: In section 3.2, the authors describe a simulation of the process of cleaning a signal from noise. In this case, white noise with an amplitude independent of frequency is used, and anti-noise processing is carried out only at high frequencies. An explanation is required on how the authors propose to combat low-frequency noise. This is especially relevant due to the fact that in a real signal the proportion of low-frequency noise may be higher.

Response 2: Agree. We have revised to emphasize this point. The low-frequency IMFs and residual terms mainly carry the long-term trends and core spectral features of the signal. They have a high correlation with the original signal and contain a large amount of useful signal information rather than dominant noise. Therefore, the framework does not perform additional denoising processing on these low-frequency components; instead, it directly retains them for the final signal reconstruction, avoiding the loss of useful low-frequency features caused by over-processing. The low-frequency components are processed in an indirect way: after the high-frequency noise is effectively suppressed by the wavelet adaptive threshold, the relative proportion of low-frequency components in the reconstructed signal increases, and their signal-to-noise ratio is indirectly improved, thereby reducing the impact of low-frequency noise on the overall signal quality.

Comments 3: I suggest writing "red dots" instead of "red line" in line 203.

Response 3: Agree. We have revised to emphasize this point. 

Comments 4: In line 216 should there be "Table 2" instead of "Table 1"?

Response 4: Agree. We have revised to emphasize this point.

Comments 5: Figure 7 is uninformative. I advise you to remove it. The whole point is reflected in Table 2.

Response 5: Agree. We have revised to emphasize this point.

Comments 6: The sensitivity and accuracy of gas analytical systems are determined by their minimum detection limit (MDL) – one of the key performance indicators. The authors of the article argue that simulation and experimental validation using the CHâ‚„ absorption spectrum at 1654 nm demonstrate that the system achieves a minimum detection limit of 0.53 ppb at 230 s integration time. This is not true. The lowest experimentally analyzed gas concentration was 10 ppm. The value of 0.53 ppb was estimated purely theoretically. The corresponding formulations in the Abstract and Conclusions need to be rewritten.

Response 6: Agree. We have revised to emphasize this point.Allan variance analysis revealed that the detection capability of the system was significantly enhanced, with the minimum detection limit (MDL) drastically reduced from 2.31 ppb to 0.53 ppb at 230 s integration time.